# Astroglial Cell-to-Cell Interaction with Autoreactive Immune Cells in Experimental Autoimmune Encephalomyelitis Involves P2X7 Receptor, β_3_-Integrin, and Connexin-43

**DOI:** 10.3390/cells12131786

**Published:** 2023-07-05

**Authors:** Katarina D. Milicevic, Danijela B. Bataveljic, Jelena J. Bogdanovic Pristov, Pavle R. Andjus, Ljiljana M. Nikolic

**Affiliations:** 1Center for Laser Microscopy, Institute of Physiology and Biochemistry “Jean Giaja”, Faculty of Biology, University of Belgrade, 11000 Belgrade, Serbia; 2Department of Life Sciences, Institute for Multidisciplinary Research, University of Belgrade, 11000 Belgrade, Serbia; 3Department of Neurophysiology, Institute for Biological Research “Siniša Stanković”, National Institute of Republic of Serbia, University of Belgrade, 11000 Belgrade, Serbia

**Keywords:** astrocytes, calcium signaling, central nervous system autoimmune disease, hemichannel, immune cell, integrin, multiple sclerosis, purinergic receptors

## Abstract

In multiple sclerosis (MS), glial cells astrocytes interact with the autoreactive immune cells that attack the central nervous system (CNS), which causes and sustains neuroinflammation. However, little is known about the direct interaction between these cells when they are in close proximity in the inflamed CNS. By using an experimental autoimmune encephalomyelitis (EAE) model of MS, we previously found that in the proximity of autoreactive CNS-infiltrated immune cells (CNS-IICs), astrocytes respond with a rapid calcium increase that is mediated by the autocrine P2X7 receptor (P2X7R) activation. We now reveal that the mechanisms regulating this direct interaction of astrocytes and CNS-IICs involve the coupling between P2X7R, connexin-43, and β_3_-integrin. We found that P2X7R and astroglial connexin-43 interact and concentrate in the immediate proximity of the CNS-IICs in EAE. P2X7R also interacts with β_3_-integrin, and the block of astroglial α_v_β_3_-integrin reduces the P2X7R-dependent calcium response of astrocytes upon encountering CNS-IICs. This interaction was dependent on astroglial mitochondrial activity, which regulated the ATP-driven P2X7R activation and facilitated the termination of the astrocytic calcium response evoked by CNS-IICs. By further defining the interactions between the CNS and the immune system, our findings provide a novel perspective toward expanding integrin-targeting therapeutic approaches for MS treatment by controlling the cell–cell interactions between astrocytes and CNS-IICs.

## 1. Introduction

Multiple sclerosis (MS) is a chronic and degenerative disease of the central nervous system (CNS) with a 30% prevalence [1]. This neuroinflammatory disease is characterized by peripheral immune cell infiltration into the CNS, oligodendrocyte loss, the formation of demyelinating lesions, astrocyte and microglial activation, that jointly cause neuronal damage [1,2,3]. Neuroimmune interactions between glial, neuronal, and immune cells play important roles in MS pathology. In MS, the infiltration of autoreactive immune cells promotes and sustains neuroinflammation, a dynamic response of glial cells to the disruption of the CNS integrity. An important part of this process is the bidirectional interaction between immune cells and glial cell astrocytes. This interaction contributes to the further infiltration of the immune cells into the CNS parenchyma and to the shaping of the astrocytic inflammatory response [4,5,6,7,8,9,10,11,12]. In addition to the communication via the release of inflammatory cues, astrocytes directly interact with the CNS-infiltrated immune cells (CNS-IICs) on a rapid time scale, indicating that such cell–cell interaction could also be a component of the neuroinflammatory response in MS [13,14,15]. However, much less is known about this direct interaction, even though this abundant type of glial cells should continuously encounter CNS-IICs in their proximity over the course of the disease. Notably, the most effective therapies for MS treatment target integrins [16], i.e., the ubiquitous transmembrane receptors that mediate cell-to-cell interaction [17], but the integrin engagement in astroglial direct interactions with CNS-IICs in MS has not been examined. 

Astrocytes create an extensive meshwork throughout the entire CNS parenchyma. These glial cells express connexins [18] that can form gap junctions which interconnect neighboring astrocytes or connexin hemichannels which provide passage for ions and small molecules such as Ca^2+^ ions, ATP, and glucose to the extracellular space [19,20,21]. Astrocytes also express a myriad of transmitter receptors and ion channels that enable communication with neighboring cells and allow them to perform maintenance functions in the CNS [22]. Receptor activation is followed by the transient increase in intracellular Ca^2+^ concentration in astrocytes [22]. Intracellular Ca^2+^ increase is caused by the entry of extracellular Ca^2+^, and/or its release from the intracellular stores such as endoplasmic reticulum and mitochondria [23]. In addition, the Ca^2+^ transporting machinery of mitochondria provides an efficient route for the regulation of cytosolic Ca^2+^ concentration [24,25]. Previously, we showed that astrocytes release ATP through connexin hemichannels upon encountering CNS-IICs, ATP then activates the P2X7 receptor (P2X7R) and triggers a Ca^2+^ increase in these glial cells [14]. However, the characteristics and the context of the astroglial P2X7R engagement that occurs in close proximity to the CNS-IICs have not been completely defined, limiting our understanding on the direct interaction between these cells in MS. 

To determine the mechanisms responsible for establishing the cell–cell interactions between astrocytes and CNS-IICs, we used the experimental autoimmune encephalomyelitis (EAE) model of MS, and we combined Western immunoblotting, immunohistochemistry, immunoprecipitation, ATP release measurement, and Ca^2+^ imaging. We show that the astrocyte response to the nearby CNS-IICs involves the coupling of astrocytic P2X7R, connexin-43 (Cx-43), and β_3_-integrin; in addition, this response is dependent on the mitochondrial activity. These data point toward the astrocyte cell–cell interactions with infiltrating immune cells as a target for the treatment of neuroinflammation in MS. 

## 2. Materials and Methods

### 2.1. Experimental Animals

The Dark Agouti (DA) rat strain was used in all experiments. Animals were housed under standard laboratory conditions in a 12 h light/dark cycle at 22–24 °C. Food and water were provided ad libitum. EAE was induced in the 2- to 3-month-old rats of both sexes, as was previously described in [14]. Briefly, spinal cord homogenate (SCH) in phosphate buffer saline (PBS; 50% *w*/*v*) was mixed with an equal volume of complete Freund’s adjuvant (CFA; Sigma-Aldrich, Darmstadt, Germany), which was supplemented with 5 mg/mL of *Mycobacterium tuberculosis* (BD Medical, Franklin Lakes, NJ, USA). Animals were intradermally injected in a hind paw pad with 100 µL of SCH and CFA emulsion. Animals were monitored daily, and the disease score was estimated based on the following scale: 0—no clinical signs, 1—flaccid tail, 2—hind limb paresis, 3—severe hind limb paralysis, and 4—limp tail, complete hind leg, and partial foreleg paralysis. Animals were sacrificed when reaching a 2–4 EAE disease score (Appendix A) together with the age-matched healthy control rats. DA rat pups (1- to 3-day-old) were used for the preparation of the spinal cord astrocyte culture. Animal procedures were carried out in accordance with the protocols of the Ethics committee for the use of Laboratory animals of the Republic of Serbia, which is in compliance with the EU directive (2010/63/EU) on the protection of animals that are used for scientific purposes.

### 2.2. Protein Isolation

Spinal cords from EAE rats (N = 8, Appendix A) and age-matched controls (N = 8) were isolated. Lumbar spinal cords were homogenized in a cold lysis buffer containing (mM) the following: 50 Tris, 150 NaCl, 10 EDTA, 10 EGTA, 0.1% SDS, and 0.5% Triton X-100. These were supplemented with protease and phosphatase inhibitors (ThermoFisher Scientific, Waltham, MA, USA), and were sonicated for 30 s. Homogenates were incubated in the lysis buffer for an additional 15 min on the ice and centrifuged at 14,000 rpm for 20 min at +4 °C. The supernatants were collected and protein concentration was determined using the Pierce Assay Kit (ThermoFisher Scientific, USA). A Laemmli sample buffer containing 1% β-mercaptoethanol was added to the samples before denaturation at 95 °C for 5 min. Samples were kept at −80 °C until needed for further use, i.e., Western blot.

### 2.3. Coimmunoprecipitation

Lumbar spinal cords were isolated from the 2- to 3-month-old DA control rats (N = 3) and rats with EAE (N = 1) (Appendix A) homogenized in an ice-cold immunoprecipitation (IP) lysis buffer composed of: 10 mM HEPES, 200 mM NaCl, 30 mM EDTA, 0.5% Triton X-100, protease and phosphatase inhibitors, and had pH 7.4. Next, the samples were clarified by centrifugation at 14,000 rpm for 20 min at +4 °C, and protein concentration was determined with a Pierce Assay Kit (ThermoFisher Scientific, USA). Samples were incubated with magnetically labeled Dynabeads^TM^ (ThermoFisher Scientific, USA) for 1.5 h, and the precleared supernatants were then used as the input for the immunoprecipitation. A total of 15 µL of magnetic beads per sample tube were incubated with a blocking solution containing 5% bovine serum albumin (BSA; Merck, Darmstadt, Germany) in an IP buffer for 1 h and 15 min on a rotator. Next, the beads were incubated with 4 µg of the rabbit anti-P2X7R primary antibody (Alomone Labs, Jerusalem, Israel) or 1 µg of the control rabbit IgG (Millipore, Burlington, MA, USA) in an IP buffer at +4 °C for 2 h on a rotator. After several washes with an IP buffer, 300 µg of the precleared input sample was added to the beads bound to the antibody and incubated at +4 °C on the rotator overnight. Bound proteins were eluted using a Laemmli sample buffer during a 15 min incubation at room temperature (RT). After the addition of 1% β-mercapthoethanol, the samples were denaturated at 95 °C for 5 min and kept at −80 °C until needed for further use, i.e., Western blot. 

### 2.4. Immunoblot

The total proteins isolated from the lumbar spinal cords of the EAE (Appendix A) and control rats, the precleared input and IP samples, or the cultured astrocytes were used for Western immunoblotting. Equal amounts of proteins (20 µg for tissue preparations and 10 µg for culture preparations) were separated by electrophoresis on 8% or 10% polyacrylamide gel. Next, the proteins were transferred to PVDF membranes (Immobilon, Merck, Germany) via a semi-dry transfer using the Trans-Blot Turbo Transfer System (Bio-Rad Laboratories, Hercules, CA, USA). PVDF membranes were next saturated in a 5% fat-free dried milk (SERVA, Heidelberg, Germany) in a Tris buffer saline with 0.1% Tween20 (TBST) (Fisher Bioreagents, Pittsburgh, PA, USA). These were then incubated overnight at +4 °C with the following primary antibodies in a 5% milk in TBST: rabbit anti-P2X7R (1:1000; Alomone Labs, Israel); rabbit β3-integrin (1:1000; Synaptic Systems, Göttingen, Germany); guinea pig anti-Cx-43 (1:1000; Alomone Labs, Israel); and rabbit anti-GAPDH (1:2000; DAKO Agilent, Carpinteria, CA, USA). The next day, the membranes were washed 3 times for 10 min in TBST and then incubated for 2 h at RT with a following horse radish peroxidase conjugated secondary antibodies: goat anti-rabbit (1:5000; Abcam, Cambridge, UK) and goat anti-guinea pig (1:5000; Santa Cruz Biotechnology, Dallas, TX, USA). After several washes, the membranes were revealed using a chemiluminescence detection kit (Western-ECL substrate, Bio-Rad Systems, Hercules, CA, USA), and were visualized using a ChemiDoc-It imager (UVP Systems, Upland, CA, USA). The experiments were performed in duplicate, and the analysis was conducted using the Gel Analyzer in Fiji ImageJ Software (NIH, Bethesda, MD, USA).

### 2.5. Spinal Cord Astrocyte Culture Preparation

1-to-3-days old DA rat pups, were used for cell culture preparation. The primary astrocyte cultures were prepared as was previously described in [14]. Briefly, the animals were decapitated, and their spinal cords were isolated. The meninges were carefully removed and the spinal cord was dissociated in PBS. The tissue was mechanically dissociated, and the homogenate was centrifuged at 500× *g* for 5 min at RT. The supernatant was discarded and the pellet was resuspended in a full Dulbecco’s modified Eagle medium (DMEM, Sigma Aldrich, Germany), which contained 10% fetal bovine serum (FBS, Gibco, Billings, MT, USA), 100 µM of sodium-pyruvate (Sigma Aldrich, Germany), and 1% penicillin-streptomycin (ThermoFisher Scientific, USA). The centrifugation–resuspension cycle was repeated, and the cell suspension was plated in a Petri dish. The cells were maintained in a humidified atmosphere of 5% CO_2_/95% air at 37 °C. The culture media was changed every 2–3 days until reaching confluence. The cells were washed with PBS, detached by trypsinization (0.025% trypsin and 0.02% EDTA in PBS) for 2 min, transferred to a tube with full DMEM, and then centrifuged at 500× *g* for 5 min at the RT. The supernatant was discarded, the pellet was resuspended in a full DMEM, and the cells were plated in a Petri dish at a density of 10^4^ cells/cm^2^. After reaching confluence, the cells were trypsinized, centrifuged at 500× *g* for 5 min at the RT, resuspended in a full DMEM, and seeded on poly-L-lysine (PLL)-coated circular glass coverslips at a density of 10^4^ cells/cm^2^. Cells were used within the following 3 days for experiments.

### 2.6. Immunofluorescence

The control rats and rats with EAE (Appendix A) were anesthetized by an intraperitoneal injection of ketamine (55 mg/kg) and xylazine (5 mg/kg), and were transcardially perfused with 0.9% physiological saline followed by 4% paraformaldehyde (PFA). The isolated spinal cords were postfixed in a 4% PFA for an additional 24 h and then dehydrated in rising sucrose concentrations of 10%, 20%, and 30% in 0.2 M phosphate buffer (PB). The tissues were frozen and cut at a 25 µm thickness using a cryostat (Leica Biosystems, Deer Park, IL, USA), and were then kept at –20 °C until needed for further use. The lumbar spinal cord sections were rehydrated in PBS for 10 min, and were then incubated for 1 h in a blocking solution containing the following: 10% normal donkey serum (NDS, Sigma-Aldrich, Germany), 2% BSA (Merck, Germany), and 0.1% Triton-X in PBS. The sections were incubated with primary antibodies in a PBS with 2% NDS at 4 °C overnight. The following primary antibodies were used: guinea pig anti-Cx-43 (1:200; Alomone Labs, Israel); mouse anti-CD4 (1:100; Abcam, UK); rabbit anti-glutamate transporter-1 (GLT-1; 1:100, Frontier Institute, Sendai, Japan); mouse anti-glutamine synthetase (GS; 1:100, BD Biosciences, Franklin Lakes, NJ, USA); and rabbit anti-P2X7R (1:200; Alomone Labs, Israel). Next, the sections were washed in PBS 3 times for 5 min and incubated with secondary antibodies in PBS with a 2% NDS at RT for 2 h. The following secondary antibodies were used: donkey anti-guinea pig Alexa 488 or Alexa 647; donkey anti-mouse Alexa 555; goat anti-mouse Alexa 488; goat anti-rabbit Alexa 555; and donkey anti-rabbit Alexa 647 (1:200 for all secondary antibodies used; Invitrogen, Waltham, MA, USA). The sections were washed in PBS 3 times for 5 min and mounted using a MOWIOL medium (Sigma-Aldrich, Germany).

For labeling the astrocytes in culture, the cells were plated on a circular glass coverslips at a density of 10^4^ cells/cm^2^. Then, they were fixed 48 h later in a 4% PFA for 30 min at RT. The cells were rinsed in PBS 3 times for 5 min, as well as blocked in a 10% NGS, 2% BSA, and 0.05% Triton-X in PBS for 45 min; they were then incubated with primary antibodies that were diluted in PBS overnight at 4 °C. The following primary antibodies were used: guinea pig anti-Cx-43 (1:200; Alomone Labs, Israel); rabbit anti-P2X7R (1:200; Alomone Labs, Israel); mouse anti-α_v_β_3_-integrin (1:100; Abcam, UK); and chicken anti-GFAP (1:500; Abcam, UK). The cells were rinsed 3 times for 10 min and then incubated for 2 h at RT with the following secondary antibodies diluted in PBS: donkey anti-guinea pig Alexa 488; goat anti-rabbit Alexa 555; goat anti-mouse AlexaFluor 555 (1:200; Invitrogen, USA); and goat anti-chicken AlexaFluor 647 (1:200; Invitrogen, USA). The cells were mounted with a MOWIOL medium after rinsing 3 times in PBS for 10 min. 

### 2.7. Confocal Image Acquisition and Analysis

Images of the immunolabeled frozen spinal cord sections were acquired on a confocal laser-scanning microscope (LSM 510, Carl Zeiss GmbH, Jena, Germany), which was equipped with Ar multi-line (488 nm) and HeNe (543 and 633 nm) lasers. The 63× (N.A. 1.4) oil immersion objective was used to acquire images with a pixel size of 143 nm in a 1 µm z-step interval. Assessment of Cx-43 and P2X7R expression and colocalization was done on 3–4 slices of the ventral lumbar spinal cord per animal. For each slice, 4 images of gray and 4 images of white matter were acquired and analyzed. Image acquisition parameters were kept constant for the control and EAE samples.

Image analysis was performed using Fiji ImageJ software (NIH, USA). The signal intensity of P2X7R and Cx-43 was measured as the integrated density. The colocalization of the Cx-43/P2X7R, Cx-43/GLT-1, and Cx-43/GS signals was assessed via Pearson’s correlation coefficient (PCC) by using the Just Another Colocalization Plugin (JACoP) [26]. The PCC was calculated from a 7–10 µm z-stack, which represented the average of the PCC values that were determined for each z-plane. The P2X7R and Cx-43 signal intensities and the distribution of their colocalization were further analyzed in manually set regions of interest (ROI): the one that covers the proximity of the infiltrated CD4^+^ T cells (tProximity of the CD4^+^ T cell), and the one that covers a random area without immune cells (Random). The analysis was performed within a 3–20 µm radius from the center of the established ROIs. The positions of the random ROIs were established within a 50 µm distance from the center of Proximity to CD4^+^ T cell’s ROIs. The distribution of the Cx-43 and P2X7R signal intensities in the established ROIs in the maximum z-projection of a 7 µm z-stack were obtained with the Radial Profile plugin. For the Sholl analysis of the Cx-43/P2X7R signal colocalizations, PCC measurements were performed in the established ROIs in a 0.5 µm radius increment of the circles. The analysis of the established ROIs excluded the signals from the labeled CD4^+^ T cells that were identified as small circles of a 5 µm radius. This enabled a specific assessment of the astrocytic P2X7R and Cx-43 signal intensities and their colocalization in the immediate proximity of the CD4^+^ T cell boundary. The number of colocalized Cx43/P2X7R signals intersecting a circle at each step size was divided with perimeter of the corresponding circle; we named this value the “density”. This analysis was also performed in a healthy animal (the control ROIs). The density of the P2X7R/Cx-43 colocalization was calculated as an average of the 7 µm z-stack, where each z-plane was analyzed separately.

### 2.8. Immune Cell Isolation

CNS-IICs were isolated form the EAE spinal cords (Appendix A) as was previously described in [27]. Briefly, the animals with EAE were anesthetized by an intraperitoneal injection of ketamine (55 mg/kg) and xylazine (5 mg/kg), and were transcardially perfused with an ice cold PBS (50–100 mL per animal). The spinal cords were isolated and mechanically dissociated in an ice-cold PBS that was supplemented with a 3% FBS (Gibco, ThermoFisher Scientific, USA). Next, the homogenate was passed through a 40 µm cell strainer to obtain the single cell suspension, which was then centrifuged at 100× *g* for 10 min at 4 °C. The supernatant was discarded and the pellet was resuspended in 3 mL of 30% isotonic Percoll (Sigma Aldrich, Germany). Next, it was slowly placed over 3 mL of 70% isotonic Percoll and centrifuged at 850× *g* for 40 min at RT. The cell layer containing CNS-IICs that were located between the 30% and 70% Percoll layers was carefully extracted and transferred to a tube with RPMI (Gibco, ThermoFisher Scientific, USA), which was supplemented with 5% FBS, centrifuged for 5 min at 500× *g* at RT, and then resuspended in an RPMI with 5% FBS. The centrifugation–resuspension step was repeated twice. The isolated CNS-IICs were resuspended in an RPMI with 5% FBS and kept at 4 °C. The CNS-IICs were centrifuged at 500× *g* for 5 min at RT, counted with a hemocytometer, and then resuspended in ECS at concentrations of 50 × or 25 × 10^3^ cells/mL prior to application.

### 2.9. ATP Release Measurement

The ATP release from astrocytes was measured using, as described in the manufacturer’s instructions, a luciferin-luciferase assay ATPLite kit (PerkinElmer, Waltham, MA, USA). The astrocytes were seeded on the bottom of the 96-well plate at a 10^4^ cells/cm^2^ density. Three experimental conditions were established: the control astrocytes in ECS, astrocytes treated with a 50 × 10^3^ CNS-IIC/well, and astrocytes that were preincubated with 20 µM CGP37157 (Tocris Bioscience, Bristol, UK) and treated with a 50 × 10^3^ CNS-IIC/well. Bioluminescence was measured using a Chameleon plate reader (Cole-Parmer, Vernon Hills, IL, USA). 

### 2.10. Time-Lapse Fluorescence Imaging

In order to monitor the Ca^2+^ dynamics in the cultured astrocytes, we used cell-permeable Fluo4-AM. The cultured astrocytes previously seeded on coverslips were loaded with 5 µM Fluo4-AM (Invitrogen, USA) for 30 min at RT in ECS. ECS contained the following (in mM): 140 NaCl, 5 KCl, 2 CaCl_2_, 2 MgCl_2_, 10 D-glucose (all from Sigma-Aldrich, Germany), and 10 HEPES (Biowest, Nuaillé, France), and had pH 7.4 and 300 mOsm osmolality. After washing 2 times for 10 min, the coverslip was placed in the recording chamber. Time-lapse imaging was carried out by using an AxioObserver A1 microscope with an LD LCI Plan-Apochromat 25× (N.A. 0.8) water immersion objective lens (Carl Zeiss), an “evolve”-EM 512 Digital Camera System (Photometrics, Tucson, AZ, USA), and VisiView^®^ high-performance software (Visi-Chrome, Visitron Systems GmbH, Puchheim, Germany). The Fluo4-AM was excited at 480 nm with a Xenon Short Arc lamp (Ushio, Tokyo, Japan), which was coupled to the VisiChrome Polychromatic Illumination System (Visitron Systems). The excitation and emission lights were passed through a FITC filter set (Chroma Technology Inc., Bellows Falls, VT, USA), and the frame scanning was performed at a frequency of 1 Hz. The cell perfusion system consisted of a three-barrel pipette connected to a Valve-Controlled Gravity Perfusion System (ALA Scientific Instruments Inc, Farmingdale, NY, USA). Each of the barrels was made of glass capillaries with an 800 µm inner diameter. The three-barrel pipette was attached to a micromanipulator to allow for the precise positioning of the pipette tip in the close vicinity of the astrocytes—specifically at the edge of the field of view, 1 mm above the coverslip surface, and at an angle of 45° relative to the bottom of the recording chamber. The accurate exchange between bathing the ECS, CNS-IICs, and ATP barrel solutions was performed using a valve-controlled system. The perfusion through the barrels was held constant at the flow rate of 4 mL/min, thus allowing for the exclusive perfusion of astrocytes via a solution from a single barrel. After 100 s of baseline recording, the bath application of ECS was switched to a solution containing CNS-IICs. The immune cells were applied for 20–30 s and then the perfusion was stopped for additional 70–80 s to allow for the establishment of a direct interaction between the CNS-IICs and astrocytes. The ECS was re-applied to a bath for 100 s to remove the CNS-IICs. Finally, the bathing solution was switched for 5 s to the one containing ATP. The CNS-IICs were applied at a concentrations of 25 or 50 × 10^3^ cells/mL. ATP was applied at 200 µM concentration (Sigma-Aldrich, Germany). ATP was applied to confirm the viability of the astrocytes.

To block the mitochondrial Na^+^/Ca^2+^ (NCLX) and H^+^/Ca^2+^ (HCX) exchangers, the astrocytes were incubated with 20 µM CGP-37157 (7-chloro-5-(2-chlorophenyl)-1,5-dihydro-4,1-benzothiazepin-2(3H)-one; Tocris Bioscience, UK). To block the astrocytic α_v_β_3_-integrin, the astrocytes were preincubated for 2 h with the primary mouse anti-α_v_β_3_-integrin antibody (1 µg/mL; Abcam, UK). The data in the presence of the drug or the antibody were compared with randomly interleaved control data that was obtained without the drug or the preincubation period. 

Analysis was performed using Fiji ImageJ software. The ROIs of the individual astrocytes were outlined based on the Ca^2+^ response evoked by ATP. The fluorescent signal intensity (F) was measured in each ROI and normalized to the baseline fluorescence (F_0_), which was conducted according to the formula of ΔF/F_0_ = (F − F_0_)/F_0_, to express the Ca^2+^ change. The astrocytes were considered responders when the CNS-IICs-evoked Ca^2+^ change was greater than 3 × SD of the baseline signal for at least 5 s. The Ca^2+^ signals detected by Fluo4-AM were quantified by measuring the area under the trace of the fluorescence change during the period of CNS-IIC presence in the proximity of the astrocytes (100 s). We integrated the consecutive ΔF/F_0_ signals as follows: 100 s before and 100 s from the start of CNS-IICs application. The resulting values are expressed as ΔF/F_0_·s in all graphs. The rate of cytoplasmic Ca^2+^ efflux was obtained by measuring the slope of the linear fit of the decay phase of the evoked Ca^2+^ response [28,29]. The area, peak amplitude, rise time, decay time, and decay slope of the Ca^2+^ change were determined in Clampfit version 10 (version 10.6.2.2. for Windows, Molecular Devices, San Jose, CA, USA).

### 2.11. Statistical Analysis

The data were analyzed and plotted using GraphPad Prism Software version 6 (version 6.01. for Windows, GraphPad Software, La Jolla, CA, USA) and SigmaPlot Software version 12 (version 12.2.0.45. for Windows, Systat Sofrware, Palo Alto, CA, USA). To assess the difference between the two groups when the data sets showed a normal distribution, a two-tailed unpaired Student’s *t*-test was used, while a non-parametric Mann–Whitney rank sum test was applied when the normality test failed. Multiple groups were compared using a one-way ANOVA analysis on the ranks (Kruskal–Wallis test), which was followed by the Student Newman–Keuls post hoc test. A two-way ANOVA analysis on the ranks was used for multiple comparisons between the group data sets. Statistical tests were used to measure the significance, along with the corresponding significance levels (*p* value) and sample size (these are shown in the figures and figure legends). The *p* values were considered significant if they were less than 0.05. All data are presented as the mean ± SEM. The dots represent individual measurements. N indicates the number of animals examined. For ATP measurement, n denotes the number of cultures from different animals. For the analysis of the time-lapse imaging data, n denotes the number of astrocytes that were obtained from at least three cultures from different animals.

## 3. Results

### 3.1. P2X7R Expression Is Diminished While Cx-43 Expression Remains Unchanged in the Spinal Cord Gray Matter in EAE

P2X7R contributes to the different pathological states of the CNS, and its modulation has been associated with the neuroinflammatory processes in EAE [30,31], as well as in patients with MS [32]. Our previous work showed that this purinergic receptor also mediates the direct interaction between CNS autoreactive immune cells and astrocytes through autocrine hemichannel-dependent astroglial signaling [14]. Hence, we first examined the expression of P2X7R alongside Cx-43 in the lumbar spinal cord of female and male rats at the peak of a severe monophasic EAE. We used Cx-43 as a marker of astrocytes as it is expressed in the plasma membrane of these glial cells in the spinal cord [33,34]. Immunohistochemical analysis confirmed Cx-43 colocalization with additional astroglial markers, such as GS and especially with GLT-1, indicating that Cx-43 is highly expressed in the plasma membrane of fine astrocyte processes (Appendix A). Our Western blot analysis revealed a prominent decrease in the P2X7R expression in the lumbar spinal cord in EAE compared to the control, while the expression of Cx-43 remained unchanged (Figure 1a–c). Furthermore, we confirmed that the expression of P2X7R and Cx-43 was similar between males and females in both the control and EAE groups (Appendix A).

Next, we investigated the expression of the P2X7R and Cx-43 expressions specifically in the white and the gray matter of the lumbar spinal cord in the EAE rats. Immunohistochemical analysis revealed that these two proteins show a diffuse expression pattern throughout the spinal cord in both, the control and EAE rats (Figure 1d,g). In the gray matter, however, the expression of P2X7R was lower in the EAE while that of Cx-43 was similar to the control (Figure 1e,f). In the white matter, on the other hand, the expression of these proteins was not altered in the EAE rats (Figure 1h,i). Taken together, these data demonstrate that in EAE, P2X7R expression is specifically diminished in the gray matter while astroglial Cx-43 expression in the spinal cord remains unaltered.

### 3.2. Astroglial Cx-43 and P2X7R Interact and Concentrate in the Close Proximity of Infiltrated CD4^+^ Cells in EAE

To further explore connection between the P2X7R and astroglial Cx-43 expression in the spinal cord, we applied the immunoprecipitation (IP) protocol using the anti-P2X7R antibody, which was coupled to magnetic beads in order to isolate the P2X7R together with its associated proteins. We tested for the presence of Cx-43 in the P2X7R immunoprecipitate complex by Western blot. The Cx-43 band was present in the input and P2X7R IP lanes, but was absent from the control IgG lane—thus showing that P2X7R interacts with Cx-43 in the spinal cord of both the control and EAE rats (Figure 2). Next, we performed a colocalization analysis of the immunofluorescence-labeled P2X7R and Cx-43 to gain further insights into the co-occurrence of these two proteins in EAE. Analysis of the gray matter revealed a decrease in the PCC between the P2X7R and Cx-43 expression in EAE rats compared to the control (Figure 3a,b). Similarly, in the white matter of the spinal cord, the PCC was diminished in EAE (Appendix A). These data show that astroglial Cx-43 interacts with P2X7R in the lumbar spinal cord, and that these two proteins are less colocalized in EAE rats. 

In EAE, the CD4^+^ T cells infiltrate into the CNS and spread through the parenchyma. Their interaction with astrocytes thus can be extracted from immunofluorescent labelings that preserve the dynamics of this interaction in the vicinity of the CD4^+^ T cells. Therefore, we further explored the distribution of the astrocytic Cx-43 and P2X7R expression in the close proximity of the infiltrated CD4^+^ cells (Figure 3c). This analysis allowed the estimation of the distribution of signal intensities along the circles of increasing radii of 3–20 µm from the CD4^+^ T cell center, whereby a 5 µm radius marked the boundary of the CD4^+^ T cell (Figure 3c). In EAE, the P2X7R signal intensity was increased in the proximity of the infiltrated CD4^+^ T cell, displaying the maximal value (6.9 ± 1.4 A.U.) at a 5.14 µm distance, which is an immediate proximity of the CD4^+^ T cell boundary (Figure 3d). 

When the same area for analysis was randomly allocated within the CNS parenchyma, the P2X7R signal intensity was significantly reduced (Figure 3d), suggesting that there is a specific localization of this purinergic receptor to the CD4^+^ T cell. This analysis was also performed for astroglial Cx-43; however, the signal intensity distribution was similar in both, the random and proximity of the CD4^+^ T cells analysis areas (Figure 3e). As in the case for the P2X7Rs, the maximal Cx-43 signal density (8.6 ± 1.0 A.U.) was found in the immediate proximity of the CD4^+^ T cell boundary, i.e., at the 5.14 µm distance. As Cx-43-mediated ATP release may activate astrocytic P2X7Rs [14], and as these proteins interact in the CNS (Figure 2), we next investigated their colocalization density relative to the CD4^+^ T cells. Our analysis showed that the signals of P2X7R and Cx-43 colocalize in the close proximity of CD4^+^ T cells (Figure 3f), and a similar association pattern of these two proteins was found in the spinal cord white matter of rats with EAE (Appendix A). Further analysis showed that the density of the colocalized P2X7R and Cx-43 signals was significantly higher in the immediate proximity of the CD4^+^ T cell boundary, which was at the 5.64 µm distance relative to the randomly chosen analysis area (Figure 3g). The colocalization of the P2X7R and Cx-43 signals was also analyzed in the spinal cord of the healthy controls (Appendix A). Furthermore, in the defined analysis areas in the gray matter, the density of the P2X7R and Cx-43 colocalized signals in the controls did not differ from the random areas in the rats with EAE (Appendix A). Collectively, these data indicate that astroglial Cx-43 and P2X7Rs interact in EAE, and that they are specifically associated and localized to the boundary of the CNS-IICs in EAE.

### 3.3. The Direct Interaction between Astrocytes and CNS-IIC Is Regulated by Mitochondria

Following the direct interaction with CNS-IICs, astrocytes release ATP in a hemichannel-dependent manner, which subsequently activates the astrocytic ionotropic P2X7R and leads to cytosolic Ca^2+^ increase [14]. The contribution of mitochondria to this astrocytic purine-based Ca^2+^ signaling, however, has not yet been determined. Indeed, in addition to their role in ATP production, mitochondria also regulate cytosolic Ca^2+^ concentration [24,25]; therefore, we blocked, using CGP37157 (20 µM), the mitochondrial Na^+^/Ca^2+^ (mNCLX) and H^+^/Ca^2+^ (HCX) exchangers that control the efflux and influx of Ca^2+^, respectively [35,36]. Previously, we found that the application of peripheral immune cells does not evoke the Ca^2+^ response of astrocytes, immune cell-derived ATP does not trigger the P2X7R activation in astrocytes, and that ATP is released by the glial cells [14]. Therefore, we first performed a luciferin-luciferase chemiluminescence assay to determine the amount of extracellular ATP released by the astrocytes in the presence of CNS-IICs following their pretreatment with CGP37157. Following the addition of CNS-IICs to the astrocytes, we observed an increase in the extracellular ATP, and this increase was more prominent when astrocytes were pretreated with the CGP37157 (Figure 4a). Next, we examined the cytosolic astroglial Ca^2+^ in response to the application of CNS-IICs and ATP in the control, as well as in CGP37157 by using the cell-permeable fluorescent Ca^2+^ indicator Fluo-4 AM (Figure 4b). Blocking astroglial mNCLX and HCX by CGP37157 induced a larger Ca^2+^ elevation in response to the bath-applied CNS-IICs compared to the interleaved control (Figure 4c,d). Notably, the astrocytes challenged with CGP37157 showed a Ca^2+^ elevation that was increased by 57% compared to the interleaved control, which is consistent with the mitochondrial contribution to the regulation of the cytosolic Ca^2+^ change that is evoked by CNS-IICs. Furthermore, the astrocytes exhibited a larger Ca^2+^ increase in response to the ATP after a block of mNCLX and HCX (Figure 4e,f), providing additional evidence for the mitochondrial involvement in regulating purine-based Ca^2+^ signals in astrocytes. Blocking mNCLX and HCX changed the decay time and decay slope of the astrocytic Ca^2+^ response that is evoked by CNS-IICs or ATP, whereas the peak amplitude and rise time remained unaltered (Figure 4g,h). With CGP37157, the cytoplasmic Ca^2+^ efflux rate, as measured from the decay slope (Figure 4g,h), was reduced by approximately 52% in the case of CNS-IICs, and 65% in case of the ATP relative to the control. These data collectively indicate that mitochondria activity shapes astrocyte Ca^2+^ signals during their interaction with CNS-IICs, and that mNCLX and HCX activity facilitates the termination of the astrocytic Ca^2+^ response that is evoked by the nearby autoreactive immune cells.

### 3.4. α_v_β_3_-Integrin Is Required for the Direct Interaction of Astrocytes with CNS-IICs and Downstream Astrocytic Signaling

Next, we investigated how the cell–cell interaction between astrocytes and CNS-IICs could be established. We focused on integrins, a ubiquitous transmembrane adhesion receptors that are composed of α- and a β-subunits with an ability to signal in both directions across the plasma membrane of the cells [16]. We specifically checked the α_v_β_3_-integrin engagement as this integrin has been associated with the astrocytic P2X7R activation by ATP released through Cx-43 hemichannels [37,38]. We first confirmed the expression of Cx-43, P2X7R, and α_v_β_3_-integrin in the spinal cord astrocytes by co-immunolabeling with GFAP (Figure 5a). Likewise, our Western blot data showed that cultured spinal cord astrocytes express Cx-43 (Figure 5b). Next, we blocked the α_v_β_3_-integrin in astrocytes with an anti-α_v_β_3_-integrin antibody (1 µg/mL), and we monitored their Ca^2+^ response to the bath-applied CNS-IICs (Figure 5c). Our data showed that the blockade of α_v_β_3_-integrin reduced the astrocyte Ca^2+^ response evoked by the CNS-IICs (Figure 5d). Notably, in the presence of the α_v_β_3_-integrin blocker, the number of astrocytes that responded to CNS-IICs (responders) decreased to 56 ± 15% compared to the 80 ± 7% of the responders in the interleaved control (Figure 5e). Moreover, when α_v_β_3_-integrin was blocked, the Ca^2+^ change in astrocytes was reduced by 31% relative to the interleaved control (Figure 5f,g). These data suggest that astroglial α_v_β_3_-integrin acts upstream of P2X7R activation and is likely involved in establishing the initial steps of direct interaction between astrocytes and CNS-IICs.

### 3.5. β3-Integrin Interacts with P2X7R and Shows Increased Expression during EAE

After we assessed the cell autonomous effect of α_v_β_3_-integrin in isolated astrocytes in culture, we next examined the association of β_3_-integrin and P2X7R in the CNS. We performed immunoprecipitation to isolate P2X7R with the associated proteins from the lumbar spinal cord of the control and EAE rats. We identified the presence of β_3_-integrin in the P2X7R immunoprecipitation complex, as demonstrated by the detection of bands in the input and IP: P2X7R lanes, as well as due to the absence of a band from the control IgG lane in both the control and EAE groups (Figure 6a). Next, we explored the expression of β_3_-integrin by Western blot in an EAE-affected spinal cord. Interestingly, our data demonstrated increased β_3_-integrin expression in the EAE lumbar spinal cord compared to the control rats (Figure 6b,c). These results indicate that β_3_-integrin interacts with P2X7R, providing further evidence for their functional coupling, as was observed here in the cultured astrocytes. Furthermore, our data on the expression analysis suggest that β_3_-integrin contributes to the pathological processes of EAE.

## 4. Discussion

A mesh-like distribution of astrocytes provides coverage of almost the entire CNS and positions them to establish multiple interactions with the CNS resident cells. These interactions occur in a healthy brain and involve the astrocytic crosstalk with neurons, microglia, oligodendrocytes, other astrocytes, and blood vessels, thus allowing astrocytes to fulfill metabolic and homeostatic maintenance functions [39,40,41]. In neuroinflammatory conditions, however, these interactions change and may affect the pathological processes that depend on the stimuli present in the inflamed milieu [42,43]. Moreover, as demonstrated in our previous study, astrocytes also establish direct cell-to-cell interactions with the CNS non-resident autoreactive immune cells that infiltrate into the CNS parenchyma [14]; extending previous reports showing that astrocyte communication with the components of the immune system is mediated by inflammatory signals. In great part, all of these cell-to-cell interactions occur through astroglial ATP-based purinergic signaling [44,45]. The signaling mediated by purines operates on a second-based time scale, enabling fast communications between astrocytes and other cells in their proximity. In a healthy brain astroglial ATP acts as a gliotransmitter, while in pathological conditions it is considered to be a danger signal. In both cases, ATP activates purinergic receptors that induce the downstream Ca^2+^ signals in astrocytes. Astroglial purinergic signaling is a hallmark of many CNS diseases, including MS [32,46]. This highlights the importance of understanding the context in which purinergic receptors are engaged and to define the mechanisms that control their signaling. By using an EAE rat model, we discovered that P2X7R interacts with astroglial Cx-43 and concentrates in the immediate proximity of CNS-IICs. In addition, we found that P2X7R interacts with β_3_-integrin and that P2X7R-dependent direct astrocyte–autoreactive immune cell interactions are dependent on astroglial mitochondrial activity and α_v_β_3_-integrins. 

Purinergic P2X7Rs contribute to the pathological events of CNS diseases such as epilepsy, Alzheimer’s disease, and MS [43,47,48]. These receptors have been connected with the profound long-lasting changes in astrocyte properties, which—in these diseases—typically induce reactive astroglial remodeling (e.g., astrogliosis) [45]. In the EAE animal model of MS, a blockade of P2X7Rs by the administration of an antagonist Brilliant blue G decreases astrogliosis in the rat forebrain and alleviates the neurological symptoms of the disease [30]. Another study using P2X7R null mice showed that P2X7R deficiency reduces EAE incidence, as well as reduces astroglial activation and axonal damage, but does not influence immune cell infiltration into the CNS [31]. MS is considered to be a white matter disease; however, lesions are also prominent in the gray matter of the brain. Accordingly, previous research demonstrated that astrocytes near the chronic active and inactive brain lesions express P2X7Rs in both the white and gray matter of the frontal cortex of patients with MS [32]. The expression of P2X7Rs was also demonstrated in the hypertrophic astrocytes surrounding the perivascular infiltrates in the brain of MS patients [46], suggesting that this purinergic receptor type is a critical component of astrocyte interaction with CNS immune cell infiltrates. By performing studies in the spinal cord of the rats with EAE, we discovered that astroglial P2X7Rs concentrate in the immediate proximity of the infiltrated CD4^+^ T cells in gray matter. We did not detect a prominent expression of P2X7Rs in astrocytes, neither in the control nor in EAE rat spinal cord tissue. This is in agreement with previous research, where it has been shown that the P2X7Rs in the CNS are localized on astrocytes at a low density [49]. In addition, aside from the analysis performed in the proximity of the CNS-infiltrated CD4^+^ T cells, Cx-43 and P2X7R showed a decrease in the colocalization in EAE. Together, these results suggest that P2X7R expression is increased and persists in the astrocytes that are in the immediate proximity of the CD4^+^ T cell in EAE, and that this receptor can contribute to the cell–cell interactions between astrocytes and autoreactive immune cells in the inflamed CNS.

P2X7R expression has been studied in MS and EAE. Augmented expression has been observed in the frontal cortex of patients with secondary progressive MS [32]. In the present study, at the peak of EAE, we observed a decrease in the P2X7R expression in the spinal cord for both females and males, while it was previously shown that the expression of this receptor increases in the forebrain [30]. Alterations in the P2X7R expression support the premise that these purinergic receptors are involved in the pathological processes of MS, however, the activity and mechanisms that control their activation remain largely unknown. Following the activation of P2X7Rs in astrocytes, many signaling pathways are activated that stimulate inflammation and also release of gliotransmitters that can modulate neuronal activity [45]. In our previous studies, we revealed that astroglial P2X7Rs are activated upon the direct astrocytic interaction with CNS-IICs [14], which could provide an explanation for the persisting expression of these purinergic receptors in the astrocytes that are in the immediate proximity of the CNS immune cell infiltrates. In the present study, we further reveal that mitochondrial activity contributes to this cell–cell interaction. The block of mNCLX and HCX mitochondrial transporters augmented the ATP release from astrocytes, and markedly increased the CNS-IICs-evoked astrocytic Ca^2+^ response by slowing the efflux of cytosolic Ca^2+^. As the activity of mNCLX is linked to oxidative phosphorylation [50], our data provide an indication of the contribution of this transporter to ATP production by astrocytes when they encounter CNS-IICs. Direct assessment, however, would help to define how mitochondrial activity is modulated during the astrocyte interaction with immune cells. In addition to producing ATP, mitochondria serve as Ca^2+^ storage organelles that efficiently regulate cytosolic Ca^2+^ [25]. This has been observed in a variety of cells including astrocytes, cardiomyocytes, and pancreatic β cells [24,51], suggesting that the mitochondrial Ca^2+^ regulation function is critically important for cellular metabolism. In acute brain slices and in vivo, mitochondria display spontaneous Ca^2+^ activity [52,53], which is thought to reflect an increase in metabolic demand and cellular stress [54]. Thus, the frequency of mitochondria-regulated Ca^2+^ transients in astrocytes increases in the animal model of amyotrophic lateral sclerosis in which the mutant enzyme superoxide dismutase is overexpressed [53]. This suggests that astrocyte “cell-intrinsic” mitochondria-regulated Ca^2+^ signals are perturbed in the CNS disease. Although further studies are necessary to assess the parallel mitochondrial and cytosolic Ca^2+^ levels in astrocytes during their interaction with CNS-IICs, our results here indicate that mitochondrial activity facilitates the termination of the CNS-IIC-evoked astroglial Ca^2+^ signals. The effects we describe appear to be mediated primarily within astrocytes since we excluded the influence of other CNS-resident cells by studying the mitochondrial regulation of the astrocyte Ca^2+^ response in pure astrocyte cultures. Notably, we observed that ATP application also evokes a larger astroglial Ca^2+^ response after blocking the mNCLX and HCX-dependent mitochondrial Ca^2+^ transports, suggesting that the regulation of astrocytic cytosolic Ca^2+^ levels by mitochondria is modified during different pathological conditions in which the level of this purine is elevated. Together, our previous [14] and herein described results suggest that mitochondria are involved in the regulation of the P2X7R activity in astrocytes. The frequent CNS-IICs-evoked P2X7R-dependent Ca^2+^ signals within astrocytes may contribute to the release of gliotransmitters and to the activation of inflammatory pathways in MS. Indeed, the block of P2X7Rs (see above) results in the reduction in astrogliosis and in the improvement of EAE symptoms. 

Essential aspects of the intracellular Ca^2+^ signals described in cultured astrocytes have been substantiated in acute brain slices, as well as in vivo [55,56,57,58]. However, the data obtained on cultured control astrocytes in this study require further confirmation through the use of brain slices and in vivo approach. This includes using animals with EAE in which mitochondria are specifically labeled, and/or the P2X7Rs are specifically removed from the astrocytes to assess the contribution of the astrocytic Ca^2+^ signaling in this disease. Notably, we also consider that the astrocyte–immune cell interaction revealed in our studies may be shaped by the inputs from other cells in the neuroinflammatory CNS environment, predominantly that derived by microglia. Indeed, on one side, microglia engage in cross-talk with CNS-infiltrated T lymphocytes and modulate immune cell function in the CNS [59,60]; while, on the other side, microglia can also promote neurotoxic astroglial activity in MS [61]. Furthermore, microglia show dynamic changes over the course of EAE, and CD4 expression in these glial cells has been correlated with the improvement of clinical symptoms for this disease [62]. Consequently, the net effect of microglial signals on the astrocyte–immune cell interaction in EAE is difficult to predict and requires detailed studies involving manipulation of different types of individual glial cell activities in different stages of disease. 

Astrocytes release ATP, which then activates the purinergic receptors through different mechanisms—including the Cx-43 hemichannels [63]. This has been shown for astrocytic P2Y1Rs in Alzheimer’s disease [64], and for astroglial P2X7Rs in our previous research on astrocyte interaction with CNS-IICs that were isolated from EAE rats [14]. Now, we reveal that Cx-43 interacts with the P2X7Rs in EAE, which provides further evidence that purine release through the hemichannels could directly activate this purinergic receptor in astrocytes. Our data also demonstrate that the association of these proteins is important for the cell–cell interactions between astrocytes and the CNS-infiltrated immune cells in EAE. Indeed, we found that Cx-43 colocalization with P2X7Rs is higher in the immediate proximity of CD4^+^ T cells. Moreover, we demonstrate that P2X7R also interacts with the β_3_-integrin in EAE, and that the block of astrocytic α_v_β_3_-integrin reduces their direct interaction with the CNS-IICs isolated from EAE rats. Previous studies showed that the functional coupling between α_v_β_3_-integrin, Cx-43, and P2X7Rs regulates astrocyte migration [38], and our data shown here further demonstrate that this coupling constitutes a signaling cascade of the events of the cell–cell interactions between astrocytes and the immune cells that invade the CNS in EAE. Moreover, our data point toward the role of mitochondria in the regulation of this coupling in astrocytes. Additional studies, however, will help elucidate the link between astroglial mitochondrial activity and α_v_β_3_- integrin in EAE. Integrins are important mediators of cell-to-cell interactions, contributing to the inflammatory processes in MS [65], and our data suggest that they could be involved in establishing a contact between astrocytes and CNS-IICs. Indeed, many integrins are upregulated in the CNS resident cells and immune cells in MS [65], and we observed an increase in β_3_-integrin expression in the spinal cord of rats with EAE. Our data on the interaction of P2X7Rs with α_v_β_3_-integrin and Cx-43, and on the expression of astrocytic P2X7Rs at the proximity to the infiltrated CD4^+^ T cells, suggest that, at a certain level, this increase in integrin expression occurs in astrocytes at the peak of EAE. Notably, integrin targeting is applied in the treatment of many pathological conditions including MS [16]. In MS, the α_4_β_7_ and α_4_β_1_-integrin blockers that are currently applied in therapeutic treatment target immune cells and control their infiltration into the CNS. Yet, immune cell-mediated mechanisms alone cannot explain the neuroinflammation and neuronal degeneration in MS without the CNS-resident cells component. Our findings thus open possibility to target astrocytes in development of innovative non-immune cell-based therapeutic strategies for treatment of MS. Such a strategy could be effective in saving neurons from damage as virtually every aspect of brain function involves a neuron–astroglial partnership.

## 5. Conclusions

The data of this study advance our understanding of the communication between the CNS and the immune system, as well as support the contribution of direct astrocyte–immune cell interactions in the pathology of EAE. Further studies will help to define the potential of astroglial P2X7R, Cx-43, and α_v_β_3_-integrin coupling in controlling the progression of EAE. This may propel the development of new astrocyte-targeting strategies for improving disease symptoms, thus preventing the formation of brain lesions and promoting the recovery of patients with MS.

## Figures and Tables

**Figure 1 cells-12-01786-f001:**
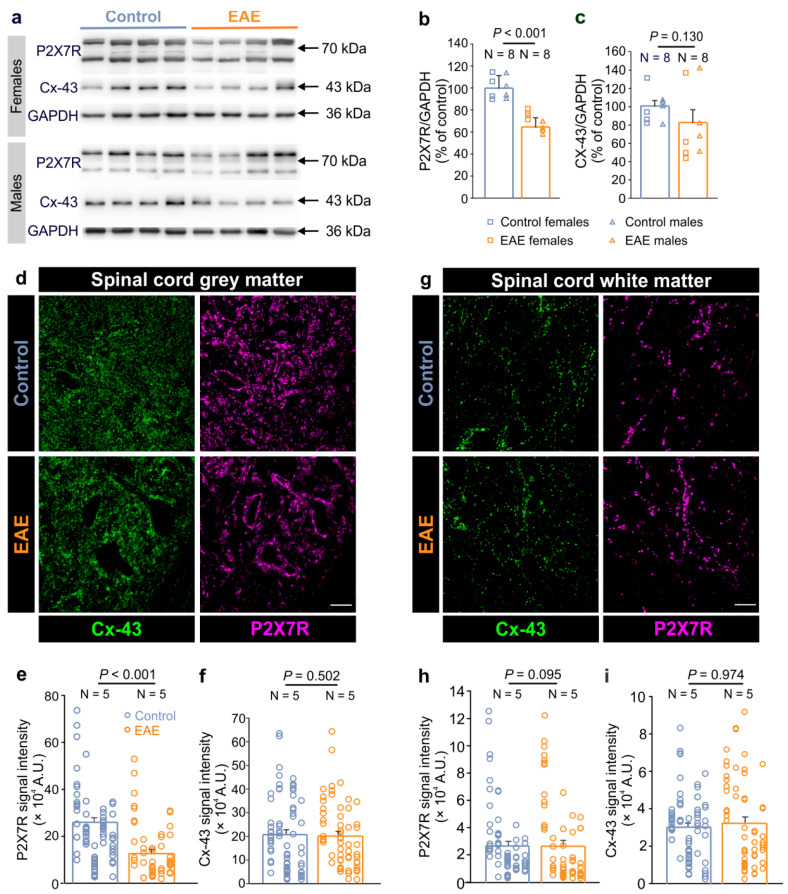
Decreased P2X7 receptor and unchanged Cx-43 expression in the spinal cord of rats with EAE. (**a**) Representative Western blots showing the P2X7 receptor (P2X7R) and connexin-43 (Cx-43) expressions in the lumbar spinal cords of the female and male control and EAE rats. GAPDH was used as a loading control. (**b**,**c**) Graphs showing the P2X7R (**b**) and Cx-43 (**c**) protein expression in the lumbar spinal cords of EAE rats at the peak of the disease (N = 8) compared to the healthy controls (N = 8) (*p* < 0.001 for P2X7R and *p* = 0.130 for Cx-43, Mann–Whitney rank sum test). The squares represent the data obtained for individual females, and triangles for individual males. (**d**) The confocal images of the lumbar spinal cord gray matter that was immunostained for Cx-43 (green) and P2X7R (magenta) in the control and EAE rats. Scale bar is 20 µm. (**e**,**f**) Graphs showing the P2X7R (**e**) and Cx-43 (**f**) signal intensity in the lumbar spinal cord gray matter of EAE rats (N = 5) compared to the controls (N = 5) (Mann–Whitney rank sum test, where *p* < 0.001 for P2X7R, and *p* = 0.502 for Cx-43). (**g**) The confocal images of the spinal cord white matter that was immunostained for Cx-43 (green) and P2X7R (magenta) in the control and EAE rats. Scale bar is 20 µm. Images in the (**d**,**g**) are maximum intensity projections of the 10 µm z-stacks. (**h**,**i**) Graphs showing the P2X7R (**h**) and Cx-43 (**i**) signal intensity in the white matter of the lumbar spinal cords in the EAE rats (N = 5) compared to the controls (N = 5) (Mann–Whitney rank sum test, where *p* = 0.095 for P2X7R, and *p* = 0.974 for Cx-43). Each vertical dot plot corresponds to the data points obtained from the individual animal. N indicates the number of animals. Data are presented as the mean ± SEM.

**Figure 2 cells-12-01786-f002:**
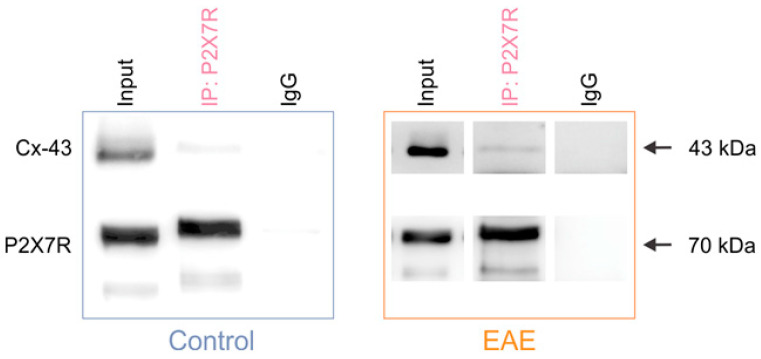
The P2X7 receptor interacts with the astroglial connexin-43 in the control and EAE rats. Representative Western blots show that Cx-43 is present in the input and P2X7R immunoprecipitate complex, and is absent from the IgG immunoprecipitate in the spinal cord of the control and EAE rats. The bands corresponding to P2X7R are identified in the input and IP lanes, but not in the IgG lane.

**Figure 3 cells-12-01786-f003:**
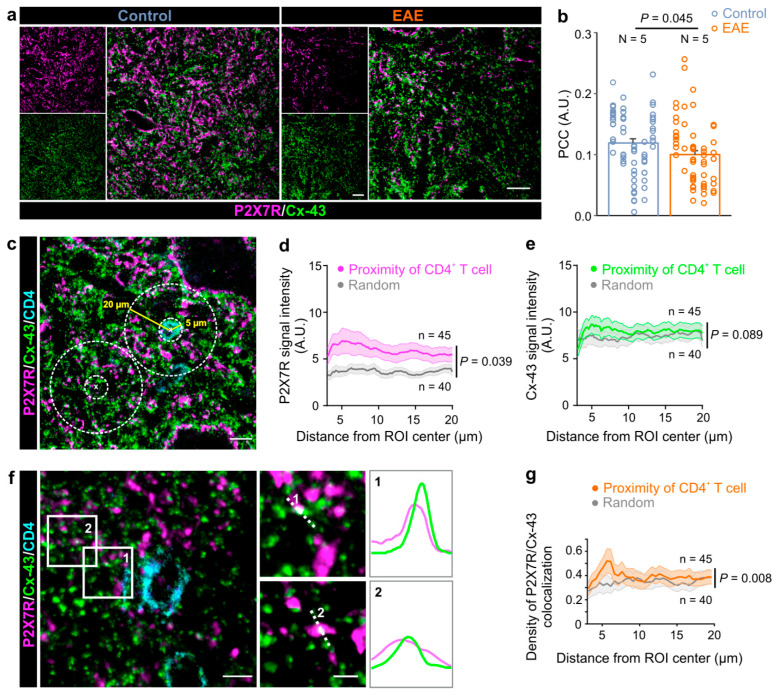
The P2X7 receptor colocalizes with the astroglial connexin-43 in the immediate proximity of CNS-infiltrated CD4^+^ T cells. (**a**) Representative confocal images of the spinal cord gray matter that were immunostained for the connexin-43 (Cx-43, green) and P2X7 receptors (P2X7R, magenta) in the control and EAE rats. Scale bars are 20 µm. (**b**) Graph showing Pearson’s correlation coefficient (PCC) of the colocalization between the Cx-43 and P2X7R expression in the spinal cord gray matter in the control and EAE rats (two-tailed Student’s *t* test, *p* = 0.045). N indicates the number of used animals. Each vertical dot plot corresponds to the data points obtained from the individual animal. Data are presented as the mean ± SEM. (**c**) Confocal images of P2X7R (magenta), Cx43 (green), and CD4^+^ T cell (cyan) immunofluorescent labeling. The depicted regions of interests (ROI) were used for analysis of P2X7R, Cx-43 signal intensity, and the colocalization in the proximity of CD4^+^ T cells and random ROIs. The yellow and dashed white lines mark the 5 µm and 20 µm radial distances, respectively, and these were measured from the center of the ROIs. Scale bar is 10 µm. (**d**) Graph showing the distribution of the P2X7R signal intensity in the proximity of CD4^+^ T cells (magenta) and random ROIs (gray) (two-way ANOVA, *p* = 0.039). (**e**) Graph showing the distribution of the Cx-43 signal intensity in the proximity of CD4^+^ T cells (green) and random ROIs (gray) (two-way ANOVA, *p* = 0.089). (**f**) Confocal images of the Cx-43 and P2X7R fluorescent signals in the vicinity of the infiltrated CD4^+^ T cells in the gray matter of the spinal cord of rats with EAE. Scale bar is 5 µm. Numbered (1, 2) white rectangles correspond to the regions presented on the right side. Scale bars are 2 µm. The profile intensity plots of the Cx-43 and P2X7R fluorescent signals were measured along each white dotted line. (**g**) Graph showing the density of P2X7R/Cx-43 colocalization in the proximity of CD4^+^ T cells (orange) and random (gray) ROIs (two-way ANOVA, *p* = 0.008). Data are presented as the mean ± SEM, and n is the number of analyzed ROIs.

**Figure 4 cells-12-01786-f004:**
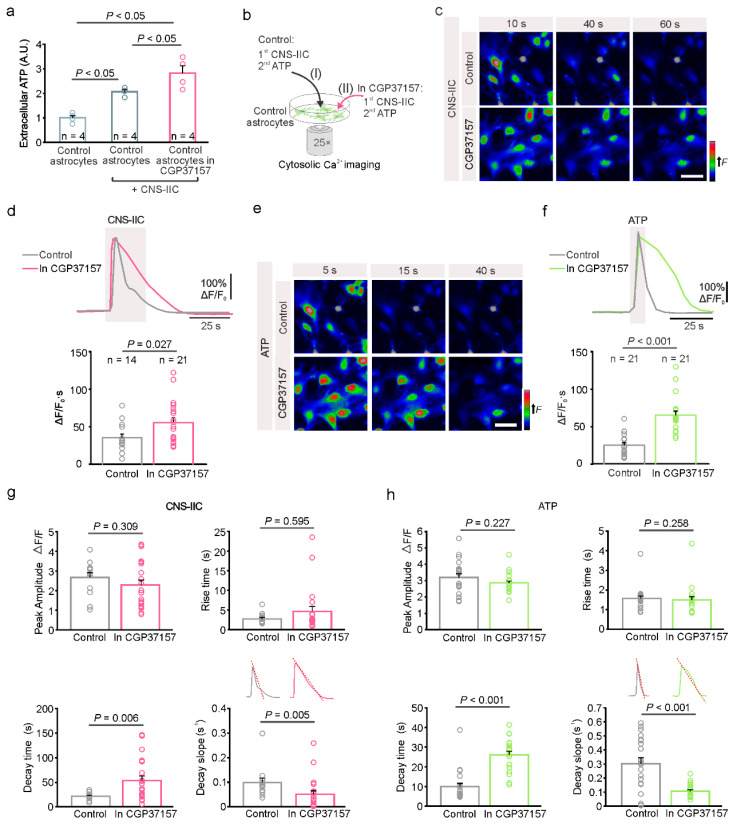
Interaction between the spinal cord astrocytes and CNS-infiltrated immune cells is mitochondria-dependent. (**a**) Summary graph of the quantitative measurement of ATP release by astrocytes when using a luciferin-luciferase bioluminescence assay. In the controls (control astrocytes) after the addition of CNS-IICs alone (control astrocytes + CNS-IICs) or in the presence of 20 µM CGP37157 (control astrocytes + CNS-IICs + CGP37157). Dots represent the individual experiments (n = 4, One-way ANOVA, *p* < 0.001; Student Newman-Keuls multiple comparisons test, where *p* < 0.05 for the controls vs. CNS-IICs, *p* < 0.05 for the controls vs. CGP37157 + CNS-IICs, and *p* < 0.05 for CNS-IICs vs. CGP37157 + CNS-IICs). The CNS-IICs were obtained from three rats with EAE. (**b**) Scheme illustrating the astroglial cytosolic Ca^2+^ imaging during the application of CNS-IICs and ATP in the control (I) and in CGP37157 (II, pretreatment with 20 µM CGP37157 for 20 min). (**c**) Colorcoded images of the Fluo 4-AM fluorescence in astrocytes during the application of CNS-IICs in the control and with CGP173157 at the indicated time points from the start of the CNS-IICs application. Scale bar is 50 µm. (**d**) Example traces and summary plot of the astrocytic Ca^2+^ increase induced by the CNS-IIC application in the control and in CGP37157 (n is the number of analyzed cells from three independent experiments, and the CNS-IICs were obtained from three rats with EAE as determined via the Mann-Whitney rank sum test, *p* = 0.027). The gray rectangle depicts the CNS-IIC application. (**e**) Color-coded images of the Fluo 4-AM fluorescence in astrocytes during a brief application of 200 µM ATP in the control and in CGP173157 at the indicated time points from the start of the ATP application. Scale bar is 50 µm. (**f**) Example traces and the summary plot of the astrocytic Ca^2+^ response evoked by ATP in the control and in CGP37157 (n indicates the number of cells from three independent experiments, Mann-Whitney rank sum test, *p* < 0.001). (**g**) Graphs showing the peak amplitude, rise time, decay time, and decay slope of the astrocytic Ca^2+^ response evoked by CNS-IICs, which was recorded in controls and in CGP37157 (Mann-Whitney rank sum test for rise *p* = 0.595, and decay time *p* = 0.006, decay slope *p* = 0.005, Student *t*-test for peak amplitude *p*= 0.309, number of cells as in (**d**)). (**h**) Graphs showing the peak amplitude, rise time, decay time, and decay slope of the astrocytic Ca^2+^ response that was induced by ATP, and recorded in the controls and in CGP37157 (Mann-Whitney rank sum test for peak amplitude *p* = 0.227, rise time *p* = 0.258, decay time *p* < 0.001, decay slope *p* < 0.001, number of cells as in (**f**)). The insets in (**g**,**h**) are as follows: the experimental paradigm for the cytoplasmic Ca^2+^ efflux rate measurement was based on the determination of the slope of the decay phase of the Ca^2+^ response. In (**d**,**f**–**h**), the dots represent the analyzed astrocytes.

**Figure 5 cells-12-01786-f005:**
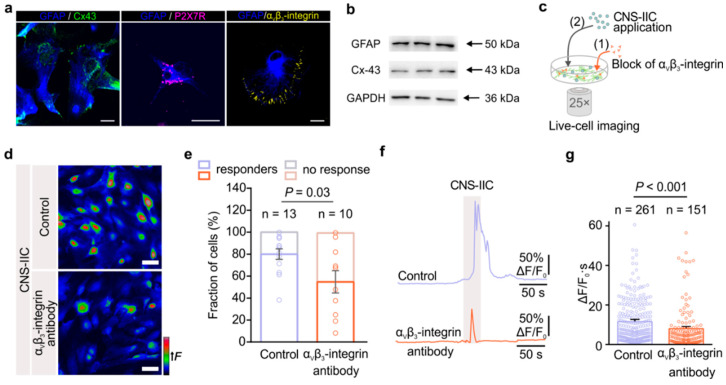
The α_v_β_3_-integrin expressed in astrocytes mediates their direct interaction with CNS-IICs and a downstream increase in intracellular Ca^2+^. (**a**) An example of connexin-43 (Cx-43, green), P2X7 receptor (P2X7R, magenta), and α_v_β_3_-integrin (yellow) immunolabeling in the control cultured astrocytes that were labeled using GFAP (blue). Scale bar is 20 µm. (**b**) Western blots showing the Cx-43 and GFAP expressions in the control cultured spinal cord astrocytes. GAPDH is used as a loading control. (**c**) Schematic depicting the imaging of astrocytic Ca^2+^ during the application of CNS-IICs in the control and after blocking the astrocytic α_v_β_3_-integrin by preincubation with a primary antibody (1 µg/mL) for 2 h. (**d**) Color-coded images of the Fluo-4 fluorescence in astrocytes during the application of CNS-IICs in the control and after blocking astroglial α_v_β_3_-integrin. Scale bar is 50 µm. (**e**) Stacked graph showing the fraction of the astrocytes that responded (responders) and that did not respond to the applied CNS-IICs (two-tailed Student’s *t* test, *p* = 0.03). Data are shown as the mean ± SEM from 13 and 10 independent experiments (depicted by dots) of the control and α_v_β_3_-integrin antibody group, respectively. N = 5 EAE rats. (**f**) Example traces showing the intracellular Ca^2+^ increase in the astrocytes following the application of CNS-IICs from the control and blocked integrin groups. Gray rectangle depicts CNS-IICs application. (**g**) Summary plot comparing Ca^2+^ elevation in the astrocytes from the experiments shown in (**d**) (Mann–Whitney rank sum test, *p* < 0.001). n is the number of responders. Dots represent the analyzed astrocytes.

**Figure 6 cells-12-01786-f006:**
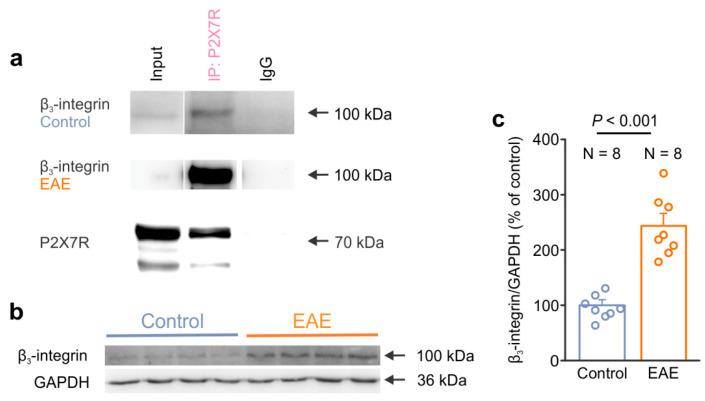
β_3_-integrin interacts with the P2X7 receptor and shows an increased expression in the spinal cord of EAE rats. (**a**) Representative Western blots showing the presence of β_3_-integrin in the P2X7R immunoprecipitate that was isolated from the lumbar spinal cord of the control and EAE rats. The bands corresponding to P2X7R were identified in the Input and IP lanes but not in the IgG lane. (**b**) Representative Western blot showing the higher expression of β_3_-integrin in the lumbar spinal cord of EAE rats compared to the control rats. GAPDH was used as a loading control. (**c**) Graph comparing the expression level of β_3_-integrin in the lumbar spinal cord of the healthy control (N = 8) and EAE rats (N = 8) (two-tailed Student’s *t* test, *p* < 0.001). Dots represent the data points of individual animals. Data are shown as the mean ± SEM.

## Data Availability

The data that support the findings of this study are available from the corresponding authors upon reasonable request.

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
