# Peer review of "Astroglial Cell-to-Cell Interaction with Autoreactive Immune Cells in Experimental Autoimmune Encephalomyelitis Involves P2X7 Receptor, β3-Integrin, and Connexin-43"

_cells, 2023, doi:10.3390/cells12131786_

Round 1
Reviewer 1 Report (Previous Reviewer 1)
General comments
We think that our demonstrations that this pathway and the coupling between P2X7R, Cx-43 and β3 integrin mediates interaction between astrocytes and autoreactive immune cells from rat with multiple sclerosis symptoms and its connection with the mitochondrial activity are sufficiently relevant and provide novel and important knowledge on the underlying pathological processes in an autoimmune disease. …..
The concept of astrocytes directly interacting with immune cells has not been anticipated and studied so far, and thus the data presented in our manuscript are of sufficient biomedical and pharmacological relevance in order to better understand pathological processes that occur in this disease.
I still believe that the novelty described in this study is not appropriate for the journal Cells, however, I understand that this decision is up to the academic editor.
We thank Reviewer 1 for helping us to strengthen our findings by bringing this up. To label astrocytes in the spinal cord we have used Cx43 as a marker, for example see the papers Scemes et al 2000 or Suadicani et al 2003.
In neither of these two articles was the CX43 immunostaining used to identify astrocytes in situ. In contrast, in Scemes et al 2000, cultured astrocytes were identified by GFAP immunostaining.
In this study we aimed to label the finest and thinnest processes of astrocytes and this is the reason why we chose Cx43 as a marker of astrocyte membrane in the fine processes instead of GFAP that would delineate only small part of the astrocyte (see Bushong et al 2002).
I agree with the authors that GFAP immunostaining does not fully portray the spongy architecture of the astrocyte network. However, I would like to note that, since 2002 considerable progress has been made in confocal microscopy techniques, and it is currently possible to provide a very detailed, though not complete, representation of astrocytic processes.
The hypothesis that CX43 immunofluorescence shown in this study is localized only on thin astrocyte processes that are not adequately marked by GFAP remains as such, because no data have been presented to confirm, or suggest it. To my opinion, it seems difficult to think that receptor clusters as conspicuous in size as those shown in confocal figures (many are abundantly larger than one micron in size), could be located on thin astrocytic processes (from the IV branching order onward). In fact, the thickness of type IV branching, as they appear by labeling for GFAP, is significantly less than 1 micron.
Figure3: The legends in figure 2 and figure 3 are reversed
Major:
As Reviewer 1 suggested, we have performed additional experiments to confirm that Cx43 is mainly expressed by astrocytes using additional astrocyte markers such as GLT1 (glutamate transporter-1) and glutamine synthase (GS). These additional experiments presented in the new Supplementary Figure 1, now show that Cx43 strongly colocalizes with GLT1, thereby confirming that Cx43 is dominantly expressed in the membrane of astrocytic fine processes in the spinal cord.
Although I do not agree with authors’ doubts on GFAP labeling, I appreciate their efforts to demonstrate that the CX43 under study is localized on astrocytes. Why did they analyze only EAE specimens? Also controls must be provided, and higher magnification details (such as in Figure 2/3 f and supplementary Fig 3b) should be added. Moreover, on a scale of 0 to 1 such as in Pearson's Correlation Coefficient, a value of 0.58 can be considered indicative of a slightly dominant, but not exclusive, localization of CX43 on astrocytes, thus again raising the same concern.
Minor
The authors responded to my technical comments satisfactorily, except for the following:
We have included XY resolution and z-step sizes in the Materials and methods paragraph ‘Confocal image acquisition and analysis’. We agree that smaller z-step intervals would bring more details; however, 1 μm was determined to be an optimal step size for optical performances of our confocal microscope that managed to reveal a higher P2X7R/Cx43 colocalisation in close proximity to the boundary of CD4+ T cell.
As far as I know, a 63x oil ommersion objective has a z-resolution of about 370nm, which in my personal experience can be reduced by deconvolution, so the authors should indicate on what basis the z-step of 1000nm was considered optimal for optical performance.
Author Response
Comments and Suggestions for Authors
General comments
We think that our demonstrations that this pathway and the coupling between P2X7R, Cx-43 and β3 integrin mediates interaction between astrocytes and autoreactive immune cells from rat with multiple sclerosis symptoms and its connection with the mitochondrial activity are sufficiently relevant and provide novel and important knowledge on the underlying pathological processes in an autoimmune disease. …..
The concept of astrocytes directly interacting with immune cells has not been anticipated and studied so far, and thus the data presented in our manuscript are of sufficient biomedical and pharmacological relevance in order to better understand pathological processes that occur in this disease.
I still believe that the novelty described in this study is not appropriate for the journal Cells, however, I understand that this decision is up to the academic editor.
We thank Reviewer 1 for helping us to strengthen our findings by bringing this up. To label astrocytes in the spinal cord we have used Cx43 as a marker, for example see the papers Scemes et al 2000 or Suadicani et al 2003.
In neither of these two articles was the CX43 immunostaining used to identify astrocytes in situ. In contrast, in Scemes et al 2000, cultured astrocytes were identified by GFAP immunostaining.
We agree with the reviewer, however, the cited papers show that genetic manipulations targeting Cx43 have impact on astroglial calcium signaling in the spinal cord and based on this we wrote that Cx43 is expressed on astrocytes. In the corrected version of the manuscript we have added Western blot data showing that Cx43 is expressed in cultured spinal cord astrocytes, to support immunohistochemistry data. We also included immunolabeling of Cx-43, P2X7 on Figure 5.
In this study we aimed to label the finest and thinnest processes of astrocytes and this is the reason why we chose Cx43 as a marker of astrocyte membrane in the fine processes instead of GFAP that would delineate only small part of the astrocyte (see Bushong et al 2002).
I agree with the authors that GFAP immunostaining does not fully portray the spongy architecture of the astrocyte network. However, I would like to note that, since 2002 considerable progress has been made in confocal microscopy techniques, and it is currently possible to provide a very detailed, though not complete, representation of astrocytic processes.
The hypothesis that CX43 immunofluorescence shown in this study is localized only on thin astrocyte processes that are not adequately marked by GFAP remains as such, because no data have been presented to confirm, or suggest it. To my opinion, it seems difficult to think that receptor clusters as conspicuous in size as those shown in confocal figures (many are abundantly larger than one micron in size), could be located on thin astrocytic processes (from the IV branching order onward). In fact, the thickness of type IV branching, as they appear by labeling for GFAP, is significantly less than 1 micron.
We acknowledge that GFAP is also appropriate marker but in our case we decided to use plasma membrane marker. This is the reason that in the corrected version of the manuscript we have added Western blot data to show that Cx43 is expressed in cultured spinal cord astrocytes, and we believe this is a good support for our immunohistochemistry data.
Figure3: The legends in figure 2 and figure 3 are reversed
We corrected this.
Major:
As Reviewer 1 suggested, we have performed additional experiments to confirm that Cx43 is mainly expressed by astrocytes using additional astrocyte markers such as GLT1 (glutamate transporter-1) and glutamine synthase (GS). These additional experiments presented in the new Supplementary Figure 1, now show that Cx43 strongly colocalizes with GLT1, thereby confirming that Cx43 is dominantly expressed in the membrane of astrocytic fine processes in the spinal cord.
Although I do not agree with authors’ doubts on GFAP labeling, I appreciate their efforts to demonstrate that the CX43 under study is localized on astrocytes. Why did they analyze only EAE specimens? Also controls must be provided, and higher magnification details (such as in Figure 2/3 f and supplementary Fig 3b) should be added. Moreover, on a scale of 0 to 1 such as in Pearson's Correlation Coefficient, a value of 0.58 can be considered indicative of a slightly dominant, but not exclusive, localization of CX43 on astrocytes, thus again raising the same concern.
We analyzed only EAE specimens since we reasoned that random ROIs that we analyzed are appropriate “internal” control to the ROIs containing CNS-infiltrated CD4+ T cells. However, as reviewer suggested we have included additional data on Control animals, please see new Supplementary figure 5. Please see previous comments regarding Cx43 expression in cultured astrocytes. We would also like to stress that in our experience with transgenic animals expression of a specific transgene in astrocytes reaches around 70% (Shen, Nikolic et al 2017), so we consider immunohistochemical colocalization value of Cx43 and GLT1 rather high.
Minor
The authors responded to my technical comments satisfactorily, except for the following:
We have included XY resolution and z-step sizes in the Materials and methods paragraph ‘Confocal image acquisition and analysis’. We agree that smaller z-step intervals would bring more details; however, 1 μm was determined to be an optimal step size for optical performances of our confocal microscope that managed to reveal a higher P2X7R/Cx43 colocalisation in close proximity to the boundary of CD4+ T cell.
As far as I know, a 63x oil ommersion objective has a z-resolution of about 370nm, which in my personal experience can be reduced by deconvolution, so the authors should indicate on what basis the z-step of 1000nm was considered optimal for optical performance.
In the materials and methods we wrote: “The 63× (N.A. 1.4) oil immersion objective was used to acquire images with pixel size of 143 nm in 1 µm z-step interval”. “Optimal step size” is not written in the manuscript and we acknowledge that it’s not correct expression. As we answered initially we agree that smaller z-step size would bring more details, however, used step size was sufficient to detect a higher P2X7R/Cx43 colocalization in close proximity to the boundary of CD4+ T cell in EAE. Furthermore, in previous corrections we have performed additional analysis as suggested by reviewer, where each optical slice was analyzed separately and similar results were obtained as when we analyzed maximum intensity z-projections. Based on this, we believe that z-step size used in our study did not compromise our analysis and obtained results.
Reviewer 2 Report (New Reviewer)
See the comments on the pdf file.

English requires an editing from a native speaker translator
Round 2
Reviewer 1 Report (Previous Reviewer 1)
There must have been a misunderstanding, I asked for immunofluorescence controls of Cx43 and GLT-1 along with related colocalization analysis, whereas the authors provided Cx43 and P2X7r immunofluorescence controls. The latter was already in the manuscript, while the former is still needed.
Author Response
We are sorry that misunderstanding occurred. Data showing Cx43/GLT1 immunolabeling in Control are now included in Supplementary Figure 2.
This manuscript is a resubmission of an earlier submission. The following is a list of the peer review reports and author responses from that submission.
Round 1
Reviewer 1 Report
The study titled "P2X7 Receptor, β3-Integrin and Connexin-43 Coupling Mediates Astroglial Cell-to-Cell Interaction with Autoreactive Immune Cells in EAE Disease" is an extension of an earlier manuscript published in 2020 by the same authors, in which it was observed that the presence of nonresident immune cells in the spinal cord of EAE animal models induces Ca2+ fluxes in astrocytes. In addition, it was highlighted that this process was dependent on the activation of astrocytic P2X7 receptors. In the current study, the authors delve into the mechanisms of P2X7r activation in these cells. Using microscopy and molecular biology techniques, it was shown that such activation exploits ATP released through Cx43 hemichannels and needs a binding between β3-integrin and P2X7r. Such interactions were described by Leyton and colleagues, in past years, and associated to the induction induce astrocyte calcium fluxes occurring in migration processes. Therefore, in novelty in this study is that the same mechanism is activated by the presence of nonresident immune cells in the EAE mouse model. The manuscript is well written and the conceptual design is appropriate. However, I believe that further investigation on what mechanisms of interaction between CNS IICs and astrocytes activate the astrocytic response, e.g. the identification of the molecule expressed by CNS IICs that interacts with astrocytic β3-Integrin in any cell-cell contacts, would have made this study of adequate interest for Cells readers. At present, to my great regret, I do not consider the results here reported to have sufficient relevance.
Here are some major comments
Figure 3: Confocal microscopy data on spinal cord preparations show the in situ association between P2X7r and Cx43, but cannot demonstrate that it occurs at the level of astrocytes, which were not labeled. In fact, both molecules are also expressed by neurons and CD4+ T cells, of which the cell body and little else is visible Therefore, to provide ex vivo representation of this association in their EAE mice, a marker revealing astrocyte processes, such as GFAP, should be also used.
Moreover, the 2D Sholl analysis counts the number of objects intersecting circles with increasing radius drawn around the ROI center. In Figure 3g, the authors show a positive correlation between the number of P2X7r/Cx43 colocalized pixels and their distance from CD4+ T cells. However, in both the previous and current studies, they conclude that the presence of CNS IICs, i.e., their contacts with astrocytes, rather than their conditioned medium, i.e., the molecules they release into the medium, showed high efficiency in inducing Ca2+ fluxes. Indeed, CNS IICs show processes that extend from the soma, but their branching during inflammation is poor. How do the authors explain this fact? Moreover, how is it possible that also in random ROIs the colocalized pixels show the same positive correlation? Given a random distribution of such interactions away from CD4 cells, the values of this ratio should be constant. Once again, the use of an astrocyte marker is probably the best method to solve these issues.
Matherials and methods
-Analyses of colocalization distribution must be performed in 3D stacks rather than in Z projections, to avoid the artifacts due to overlapping of signals that are not colocalized in the Z-axis.
-Original XY resolution and Z-step size of all 3D confocal scans used for intensity as well as colocalization analyses must be given. Moreover, 1 μm z-step intervals is too high to correctly examine colocalization.
-”We used 63X objective (N.A. 1.4) and additional 4 x zoom”, authors must clarify how the 4 x Zoom was obtained. To my opinion, if the NA of the objective remains unchanged the same goes for the resolution of the acquisition, which does not change. Any further digital zoom would not help to increase the resolution.
Reviewer 2 Report
This study by Milicevic et al tests the effect of CNS inflitrated immune cells on P2X7 receptor activation and calcium influx in astrocytes. The authors suggest that coupling between P2X7R, connexin-43 and beta3-integrin in astrocytes is necessary for this process to occur. They also suggest that mitochondrial activity in astrocytes regulates ATP-mediated P2X7 activation, and this interaction is important in the context of EAE. Overall, this is an interesting study, however there are major and minor points that will need to be addressed for the authors to experimentally corroborate their conclusions. These are listed below:
(1) A major problem is are data shown in Figures 4 and 5, which is central to the conclusion of this paper. The authors have not considered the idea that ANS-IIC can release ATP and alter the amount of extracellular ATP as well as the activity of astrocytic mitochondria by directly activating P2X7 receptors. An important control to assess this idea is to measure extracellular ATP and with only CNS-IIC. It is also not clear how the cells are rapidly perfused and removed from the bath. Methods need to describe this is detail.
(2) Another major problem is that the authors state the idea that increased calcium influx into astrocytic mitochondria would reduce ATP production. This is very confusing. Calcium influx into mitochondria activates calcium-dependent dehydrogenases to produce NADP which is an important co-factor for the production of ATP by cells. The authors need to show that reduced calcium influx into astrocytic mitochondria increases extracellular ATP by measuring extracellular ATP in the presence of the alphavbeta3 integrin antibody as shown in Figure 5e. Findings in this regard could likely be an artefact of doing the experiments in cultured astrocytes. Therefore, these experiments will need to be done in live brain slices in order to make more relevant conclusions. It is well known that astrocytic mitochondria in situ demonstrate spontaneous calcium influx with astrocytes cultured in a dish do not do so, which suggests that the findings related to calcium influx / efflux, ATP generation and P2X activation are likely not physiological. This major limitation of the study and brings into question the authors conclusions.
(4) It is appropriate to cite Huntington et al, 2021 (Cell Calcium), which shows that astrocytic mitochondria in brain slices show spontaneous calcium events. This can be done in the introduction and / or discussion of the paper.
(5) There are several minor typos grammatical syntax errors and convoluted sentences in the writing that need to be corrected and proofread carefully.
(6) Please expand the abbreviation EAE to full form in the title.